# A molecular insight into the lipid changes of pig *Longissimus thoracis* muscle following dietary supplementation with functional ingredients

**Gabriele Rocchetti**[1☯‡], **Marika Vitali**[2,3☯‡], **Martina Zappaterra**[3]*, **Laura Righetti**[4]*, **Rubina Sirri**[2,3], **Luigi Lucini**[5], **Chiara Dall'Asta**[4], **Roberta Davoli**[2,3], **Gianni Galaverna**[4]

**1** Department of Animal Science, Food and Nutrition, Università Cattolica del Sacro Cuore, Piacenza, Italy, **2** Interdepartmental centre for Industrial Agrifood research (CIRI-AGRO)—Università di Bologna, Cesena, Italy, **3** Department of Agricultural and Food sciences (DISTAL), Alma Mater Studiorum–Università di Bologna, Bologna, Italy, **4** Department of Food and Drug, Parco Area delle Scienze, Parma, Italy, **5** Department for Sustainable Food Process, Università Cattolica del Sacro Cuore, Piacenza, Italy

☯ These authors contributed equally to this work.
‡ GR and MV are the co-first authors.
* martina.zappaterra2@unibo.it (MZ); laura.righetti@unipr.it (LR)

**Data Availability Statement:** All relevant data are within the manuscript and its Supporting Information files.

## Abstract

In this work, the *Longissimus thoracis* pig skeletal muscle was used as a model to investigate the impact of two different diets, supplemented with *n-3* polyunsaturated fatty acids from extruded linseed (L) and polyphenols from grape skin and oregano extracts (L+P), on the lipidomic profile of meat. A standard diet for growing-finishing pigs (CTRL) was used as a control. Changes in lipids profile were investigated through an untargeted lipidomics and transcriptomics combined investigation. The lipidomics identified 1507 compounds, with 195 compounds fitting with the MS/MS spectra of LipidBlast database. When compared with the CTRL group, the L+P diet significantly increased 15 glycerophospholipids and 8 sphingolipids, while the L diet determined a marked up-accumulation of glycerolipids. According to the correlations outlined between discriminant lipids and genes, the L diet may act preventing adipogenesis and the related inflammation processes, while the L+P diet promoted the expression of genes involved in lipids' biosynthesis and adipogenic extracellular matrix formation and functioning.

## Introduction

Using nutritional strategies to improve the overall quality of meat and meat-derived products is an emerging approach, representing a bridge between food and animal sciences [1, 2]. The most used strategies were based on changes in the nutritional profile by supplementing animal diets rich in polyunsaturated fatty acid (PUFA) with antioxidant compounds (such as vitamin E) to improve the oxidative stability of the meat [3]. In this regard, following supplementation with a PUFA source, it is important to add also natural antioxidants that minimize pigment and lipid

**Funding:** This research was funded by Regione Emilia-Romagna POR-FESR 2014-2020(URL: https://fesr.regione.emilia-romagna.it/) "Innovare la filiera suina mediante la valorizzazione di sottoprodotti vegetali e l'impiego di avanzate tecnologie "omiche"e di processo, per la produzione sostenibile di carne e salumi ad impatto positivo sulla salute"-GreenCharcuterie grantn.PG/2015/730542.The funders had no role in study design, data collection and analysis, decision to publish, or preparation of the manuscript. The funding was received by RD and GG.

**Competing interests:** The authors have declared that no competing interests exist.

oxidation phenomena [4]. In recent years, food producers and consumers have shown an increasing interest in the origin of feed additives, and particular attention has been devoted to natural compounds, such as polyphenols. These latter are recognized as antioxidants and their effectiveness is comparable to that of vitamin E [5]. Therefore, they are potentially able to prevent the oxidative deterioration of lipids (such as PUFA) and proteins of meat [6].

The modification of lipid profile is related to the fact that fatty acids are strictly associated with meat quality, sensory attributes, and meat nutritional value [7]. High PUFA levels may alter meat flavor because of their increased susceptibility to oxidation, thus leading to the production of off-flavors during the cooking process. Also, from a nutritional standpoint, the interest towards increasing PUFA in meat depends on the demonization of saturated fatty acids (SFA), previously described to be associated with cardiovascular diseases and cancer [8]. Regarding meat quality, the variation of fatty acid composition can affect some parameters, such as the firmness and oiliness of adipose tissue and the oxidative stability of muscle. This latter parameter is also correlated to potential changes of both flavor and muscle colour [7].

Therefore, starting from the previous considerations, meat producers are attempting to improve the nutritional quality of meat by adding different sources of *n-3* PUFA in the animal diet, considering that fatty acids are absorbed from the diet and incorporated into tissue fat [9]. The major dietary sources of *n-3* fatty acids in swine nutrition are represented by vegetable oils, fish oil, and fish meal. Also, one of the most used supplementation strategies is based on linseeds, representing a rich source of α-linolenic acid. The utilization of linseeds is correlated to increasing levels of *n-3* PUFA in tissue and a corresponding decrease of the atherogenic and thrombogenic index of pork [10]. However, it is also important to highlight that gilts and barrows respond differently to linseed supplementation due to a different ability to store fat in the body, a mechanism mainly regulated by sex hormones [8]. Additionally, as suggested by Flachowsky and co-authors [11], the fatty acid composition of backfat and intramuscular fat showed much higher differences between fat supplements to the diets than between sexes. In previous works from our research group [4, 12], a transcriptomic approach revealed that the supplementation of pig diet with *n-3* PUFA derived from linseed, antioxidant molecules and plant polyphenols can affect the expression of a high number of genes in *Longissimus thoracis* muscle. Those genes were mainly involved in biochemical pathways related to muscle function (such as development and physiology) and lipid biosynthesis. However, to the best of our knowledge, no studies are available in the scientific literature on the comprehensive changes of the lipidomic profile of pigs fed with different functional diets.

Starting from the previous considerations, this work aimed to comprehensively investigate (for the first time in pig skeletal muscle) the impact of two different diets based on the supplementation with extruded linseeds (a natural source of *n-3* PUFA) and plant extract (a source of antioxidant polyphenols) on the lipidomic profile of pork. Therefore, UHPLC-QTOF mass spectrometry was coupled with robust univariate and multivariate statistical approaches to reach this goal. Also, the expression levels of genes involved in both lipid and energy metabolism, together with fat deposition, were tested in the *Longissimus thoracis et lumborum* muscle samples of Italian Large White (ILW) pigs. Our findings are expected to enlarge the limited information available on the correlation existing between the lipidomic profile of meat and gene expression modulation characterizing the different experimental diets.

## Materials and methods

### Chemicals and reagents

Polytetrafluoroethylene (PTFE) 15 mL centrifugation cuvettes were obtained from Greiner Bio-One (Kremsmünster, Austria). HPLC grade methanol, ethanol, dichloromethane,

2-propanol, and *n*-hexane were purchased from Merck (Darmstadt, Germany). Ammonium formate and formic acid were supplied by Sigma–Aldrich (St. Luis, MO, USA). Water was purified by Milli-Q purification system (Millipore, Bedford, MA, USA).

## Ethics approval

The present study uses samples that were already included in previous studies [4, 12]. In particular, pigs were reared following the Council Directives 98/58/EC and 2008/120/EC, and animal transport and slaughtering were performed according to Council Regulations (EC) N. 1/ 2005 and N. 1099/2009. Slaughter was performed at a commercial abattoir under the control of the Veterinary Service from the Italian Ministry of Health. The studies were conceived with the Council for Agriculture and Agricultural Economy Analysis partner, a member of the National Institutional Animal Care Committee, which issued in the decision included in the Report 2 of 2016 September, the 14th that all the procedures performed in this study complied with the Italian legislation, D.Lgs 4 Marzo 2014 n. 26 art. 2 punto F.

## Animals, diets, and sampling

The samples used in the present study were gathered from 36 ILW pigs (18 gilts and 18 barrows), already described in our previous transcriptomic studies [4, 12]. After the pigs reached about 80 kg live weight, they were assigned to three diets (12 pigs for each diet): a standard growing-finishing diet (CTRL); a diet enriched in extruded linseed (5% of feed on wet basis; L); a diet supplemented in extruded linseed and plant extracts obtained from grape-skin (3 g/ kg feed; Enocianina Fornaciari s.n.c., Reggio Emilia, Italy) and oregano (2 g/kg feed; Phenbiox Srl, Bologna, Italy; L+P). The latter diet (L+P) was conceived to test the combined effect of *n*-3 PUFAs (supplied with the extruded linseed) and polyphenols (total polyphenols added 37.6 mg per kg of feed). Pigs were fed the diets at 7.5% of the metabolic live weight per day during the first period (lasting from 79.9 ± 5.8 kg to 113.4 ± 10.6 kg), and then at 8.5% of the metabolic live weight (period lasting from 113.4 ± 10.6 kg to slaughter at 150.5 ± 9.9 kg live weight). S1 Table reports the detailed description and the nutritional contents of the three diets. During the trial, one barrow fed L+P died because of an abdominal hernia, and the L+P group was therefore composed of 11 replicates. The trial ended when pigs reached an average weight of about 150 kg. Pigs were all slaughtered in the same commercial abattoir. At the end of the slaughter line and before carcasses were cooled down, two samples of the *Longissimus thoracis et lumborum* muscle were gathered from the left side of each carcass, on the back over the joint between the last thoracic and first lumbar vertebrae. The two samples were immediately frozen in liquid nitrogen, and stored at -80˚C until further analysis.

## Sample preparation and untargeted UHPLC-QTOF lipidomics analysis

In this work, 1 g of grounded fresh pork was extracted in a 10 mL mixture of dichloromethane/methanol (50/50, v/v), by using an automatic shaker (IKA Laboratortechnik, Germany) for 30 min at 240 strokes/min. Each extract was then centrifuged for 5 min at 20˚C, at a speed of 13,416 x g (Rotina 35 R, Hettich Zentrifugen, Germany). Finally, 1 mL of each extract was evaporated under nitrogen and the residue reconstituted to the same volume with 2-propanol/methanol/water (65:30:5, v/v/v).

The chromatographic conditions were optimized in our previous works [13, 14]. Briefly, a 1290 UHPLC system (Agilent Technologies, Santa Clara, CA, USA) was equipped with a BEH $C_{18}$ (2.1x100 mm, 1.7 μm) column thermostated at 60˚C. The mobile phases consisted of (A) 5 mM ammonium formate and 0.1% formic acid in water/methanol (95/5, v/v), and (B) 5 mM ammonium formate and 0.1% formic acid in 2-propanol/methanol/water (65/30/5, v/v/v). A

multi-step elution dual-mode gradient (total run: 17.5 min) was previously optimized [13]. The sample injection volume was 1 μL and the autosampler temperature was kept at 5°C. Regarding the untargeted High Resolution Mass Spectrometry (HRMS) analysis, a 6550 iFunnel QTOF mass spectrometer (Agilent Technologies, Santa Clara, CA, USA) was used. The mass spectrometer worked in both positive and negative ionization modes, considering the acquisition range 100–1200 *m/z* at a rate of 0.8 spectra/s (absolute peak height threshold 3000 counts, relative height threshold 0.0001%) and the extended dynamic range mode (with a nominal mass resolution of 30,000 FWHM). The electrospray ionization (ESI) ion source parameters were optimized in previous works [14]. Also, quality control (QC) samples were prepared by pooling an aliquot of the different extracts, to be analysed using a data-dependent MS/MS method. The method consisted of an MS survey ranging from *m/z* 100 to 1200, followed by MS/MS acquisition in the range from m/z 50 to 1200, for the top N = 10. In-batch samples were analyzed randomly (established based on random number generation) to avoid any possible time-dependent change during UHPLC-QTOF analysis, potentially resulting in false clustering. Each set of samples was preceded by 3 blank controls: Milli-Q water, methanol, and blank (extraction procedure without sample). To address overall process variability, metabolomics studies were augmented to include a set of eighteen (20% of the entire sample set) technical replicates and pooled quality control. According to this approach, repeatability, reproducibility, precision, and mass accuracy of metabolites were preserved. Finally, the raw spectral data were processed using the software MS-Dial (version 4.20) [15], following the "Lipidomics" workflow, based on both LipidBlast annotations and theoretical MSMS spectra for lipids internally included in the software.

## Data processing and multivariate statistical analyses of UHPLC-QTOF data

The deconvolution process of the annotated raw mass features was done using the software Profinder B.07 (from Agilent Technologies). Data pre-processing (mass and retention time alignment, features filtering) was conducted in Profinder B.07. The mass features were filtered by abundance and frequency (only those compounds with an area > 10000 counts and appearing in 80% of samples in at least one condition were considered), normalized at the 75th percentile and baselined to the median of each compound in all samples. The generated dataset was then exported in SIMCA 13 software (Umetrics, Malmo, Sweden), Pareto scaled, and the presence of outliers was evaluated according to Hotelling's T2 distribution. The quality of the prediction models was evaluated by inspecting the goodness-of-fit ($R^2X$), the proportion of the variance of the response variable that is explained by the model ($R^2Y$) and the predictive ability parameter ($Q^2$), which was calculated by seven-round internal cross-validation of the data using a default option of the SIMCA software. Also, to avoid the risk of overfitting, statistical models were cross-validated, according to the leave one-third out approach.

Thereafter, the multivariate statistical analysis on the metabolomics dataset provided by MS-Dial and MS-Finder software was carried out using the online tool MetaboAnalyst [16]. Briefly, MSMS annotations were normalized by the median, log-transformed, and Pareto scaled. After that, a Volcano Plot analysis was done by coupling one-way ANOVA ($p < 0.05$; FDR-adjustment) and Fold-Change analysis (cut-off > 1.2). Also, both unsupervised (i.e., principal component analysis- PCA) and supervised (PLS-DA and OPLS-DA) statistical models were built to check the overall variability explained by the metabolomics dataset, together with checking the most discriminant metabolites for the comparison L *vs.* CTRL and L+P *vs.* CTRL, respectively. In this regard, we considered significant those features presenting a VIP (variable importance in projection) score > 0.8, according to our previous works [17]. Also, the S-plot

was built in the software MetaboAnalyst to combine the modelled covariance and correlation from the OPLS-DA models. Besides, a chemical enrichment approach (ChemRICH) [18] was used considering both comparisons (i.e., L *vs.* CTRL and L+P *vs.* CTRL). ChemRICH is based on structure similarity and chemical ontologies to map all known metabolites and identify the most represented metabolic modules. Unlike pathway mapping analysis, this strategy yields study-specific, non-overlapping sets of all identified metabolites. To discern among the up-accumulated marker compounds specifically related to L and L+P diets vs the CTRL diet, a Venn diagram was carried out using an online tool (Venny 2.1 - http://bioinfogp.cnb.csic.es/tools/venny/index.html). Finally, a sparse (s)-PLS-DA score plot together with a VIP selection method was inspected to provide the impact of polyphenols provided by grape skin and oregano extracts on the lipidomic profile detected (i.e., comparing L vs L+P experimental diets).

## Correlations between differentially expressed genes and discriminant lipid classes

The list of the differentially expressed genes (DEGs) in L *vs.* CTRL, L+P *vs.* CTRL, and L +P comparisons were retrieved from our previous studies [4, 12]. To increase the knowledge on the correlation between the lipidomic profile and gene expression in the meat of pigs fed different diets, a statistical analysis was performed between the DEGs and the discriminant lipids found with OPLS-DA. Due to the large amount of data to deal with, we decided to manage separately the information obtained from the comparisons L *vs.* CTRL, L+P *vs.* CTRL, and L *vs.* L+P. First, a cluster analysis was performed to explore if the expression and accumulation level of genes and lipids could discriminate between the two diets in each comparison. The optimal number of clusters was chosen applying both elbow and silhouette methods [19]. Then a PCA was conducted in each comparison to explore which of the DEGs and lipids contributed the most to the total dataset variability. PCA is based on the variance-covariance matrix of the variables submitted to the analysis and captures the total variability of the dataset into a new set of variables (Principal components, PCs) originating from linear combinations of the original variables in the dataset. Each sample is given a value (loading) for each PC; loadings are therefore the coordinates of the samples in the PCA multidimensional space and indicate how the features of each sample contribute to explaining the total variability of the dataset. Thus, PCA was used in the present study to identify the multivariate relations linking the DEGs and lipids in each comparison between treatments. After PCA was run, PC loadings were extracted and used to perform a correlation analysis with the original variables (i.e., the expression of DEGs and the accumulation levels of the lipids). This approach was applied following Aboagye et al. [20] to identify the compounds and genes contributing the most to the new PCs. The genes and compounds correlated with PC loadings with an absolute value of the correlation coefficient ($|r|$) > 0.60 and a *p*-value < 0.050 were considered for further investigations. Those genes and lipids were indeed submitted to a new correlation analysis to find which variables were the most correlated and thus highlight possible molecular and metabolic patterns relating the genes and compounds that were significantly altered by the diets L and L +P. All analyses described in the present paragraph were performed in the R environment (https://www.r-project.org/). The cluster analysis was performed using *cluster* and *factoextra* packages, and the packages *stats* and *Hmisc* were used to perform the PCA and Pearson correlations, respectively.

## Results and discussion

This study was carried out as a part of a wider research project that was aimed at studying the effects of diets enriched in extruded linseed and antioxidants on the carcass and meat quality

features of medium-heavy pigs. The experimental diets did not influence the Average Daily Gain, nor the slaughter live weight of the animals, while the pigs fed L+P diet showed the best Feed Conversion Ratio (L+P = 3.47 kg/kg *vs*. CTRL = 3.81 kg/kg, *p*-value < 0.05) [21]. Furthermore, no differences were identified with regard to carcass composition, pH, and lipid oxidation in fresh and cooked meat [21]. Total lipids in *Longissimus thoracis* muscle were not statistically different, nor the total SFA and monounsaturated fatty acids, while *n*-3 PUFA were significantly higher in the *Longissimus thoracis* muscle of pigs fed L (3.55% on total fatty acids) and L+P (3.31% on total fatty acids) compared with CTRL group (1.25% on total fatty acids; *p*-value < 0.05) [21].

## Multivariate statistical discrimination of the annotated mass features

In this work, an untargeted lipidomics-based approach was used as a key tool to identify the lipid sub-classes being modified by the different feed regimens under investigation. The final aim was to correlate the discriminant lipids with specific physiological processes. As a first evaluation, the unsupervised multivariate statistics based on principal-components analysis (PCA) were carried out, using the UHPLC-QTOF mass features to detect possible sample grouping trends, thus assessing the scattering within treatments. Besides, the 47 chromatograms obtained in positive and negative ionization modes (S1 Fig) were separately submitted for multivariate and univariate analysis. The metabolomic dataset from the molecular-feature-extraction analysis showed 2912 and 436 features for positive and negative ionization modes, respectively. The presence of natural grouping within the dataset was observed in the unsupervised models (i.e., PCA), as depicted in Fig 1. The control diet samples (CTRL) were separated from the other two diets (L and L+P), which were found to cluster together in the PCA score

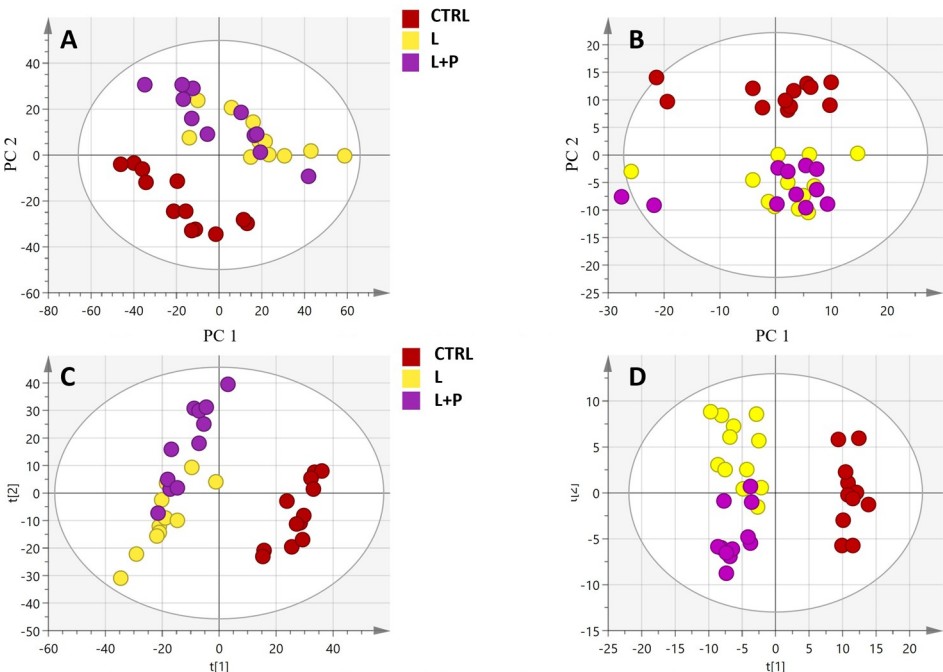

**Fig 1.  Unsupervised principal components analysis (PCA) models built using raw data from positive (A; $R^2$cum = 0.39) and negative (B; $R^2$cum = 0.59) ionization modes.** The diet groups i.e., control (CTRL), linseeds (L), linseeds and polyphenols (L+P) are color-coded accordingly. Also, the supervised OPLS-DA models for positive (C; $R^2$cum = 0.99, $Q^2$cum = 0.74) and negative (D; $R^2$cum = 0.75, $Q^2$cum = 0.34) modes are reported.

plot hyperspace, when combining ESI(+) and ESI(-) molecular features (Fig 1A and 1B). Interestingly, no grouping could be observed according to the sex and slaughter time, thus indicating a hierarchically major role of the different feeding regimens. The clustering became evident when a supervised OPLS-DA multivariate approach was adopted (Fig 1C and 1D). Thereafter, additional PCA models were built using each time two groups in order to maximize the between-groups differences, comparing CTRL *vs*. L and CTRL *vs*. L+P diets. A tendency to differentiate the treatments according to metabolic profile could be observed from the unsupervised models that are summarized in S2 and S3 Figs. Furthermore, following an L *vs*. L+P comparison, no particular clustering could be observed (S4 Fig).

## Discrimination between CTRL and L+P diets based on lipidomic profiling

Starting from the differences previously observed by supervised and unsupervised statistical models built on the MFs, the software MS-Dial 4.20 was used to annotate the most important lipid classes driving the discrimination between CTRL and L+P diets. Overall, this approach allowed us to annotate a total of 1507 compounds, with 195 compounds fitting against the MS/MS spectra reported on LipidBlast. The remaining 1312 compounds were annotated according to a Level 2 of confidence, as set out by the COSMOS Metabolomics Initiative [22] (i.e., putatively annotated compounds). A comprehensive table reporting the retention time, mass, adduct type, formula, MS1 isotopic spectrum, MS/MS spectrum, together with other annotation parameters of each metabolite identified in both positive and negative ESI polarities, can be found in S2 Table.

The most represented lipid subclasses in ESI (+) were those of triacylglycerols (273 compounds), followed by phosphatidylcholines (160 compounds), sulfolipids (157 compounds), sphingolipids (such as sphingomyelins; 112 compounds), and diacylglycerols (90 compounds). On the other hand, the ESI (-) acquisition revealed a lower number of compounds (280 annotations), with a great abundance of glycerophospholipids, such as ether-linked phosphatidylethanolamines (29 compounds) and ether-linked phosphatidylcholines (24 compounds), followed by sphingomyelins (19 compounds). As previously reported in scientific literature, the fatty acid profiles of meat can be affected by several factors, including the genetic background [23], dietary composition [24], slaughter age [25], and production systems [26]. However, although several papers have been published about the fatty acid composition of pork muscles [23, 27, 28], a more comprehensive lipidomic profile of pork, when considering different feeding systems, is still lacking. Overall, the number and typology of lipid annotated were in agreement with previous works on pork meat. In this regard, Mi and co-authors [29], using an LC-MS-based lipidomic profiling approach, discriminated selected China's domestic pork, showing a great abundance of glycerophospholipids (38.31%), followed by glycerolipids (20.51%), fatty acyls (15%), sphingolipids (9.49%), and some other minor subclasses.

After that, we combined univariate and multivariate statistical approaches in order to extrapolate those lipid marker compounds showing significant differences ($p < 0.05$) between the two different groups (L+P *vs*. CTRL). Overall, 154 lipids were found to possess the highest discrimination potential, as provided by Volcano plot analysis (i.e., combining ANOVA and Fold-Change analysis), supervised OPLS-DA followed by VIP selection method, and S-plot graphs. The OPLS-DA score plot, together with Volcano and PCA score plots, are reported in S2 Table. Besides, a comprehensive picture of the most discriminant compounds is also provided in S2 Table, reporting the chemical subclass, together with the formula, adduct type (in both ESI ionization modes), annotation level, and accumulation trends (i.e., up or down) when considering the comparison L+P *vs*. CTRL. As can be observed from S2 Table, the most represented lipid subclasses included triacylglycerols (32 compounds), followed by

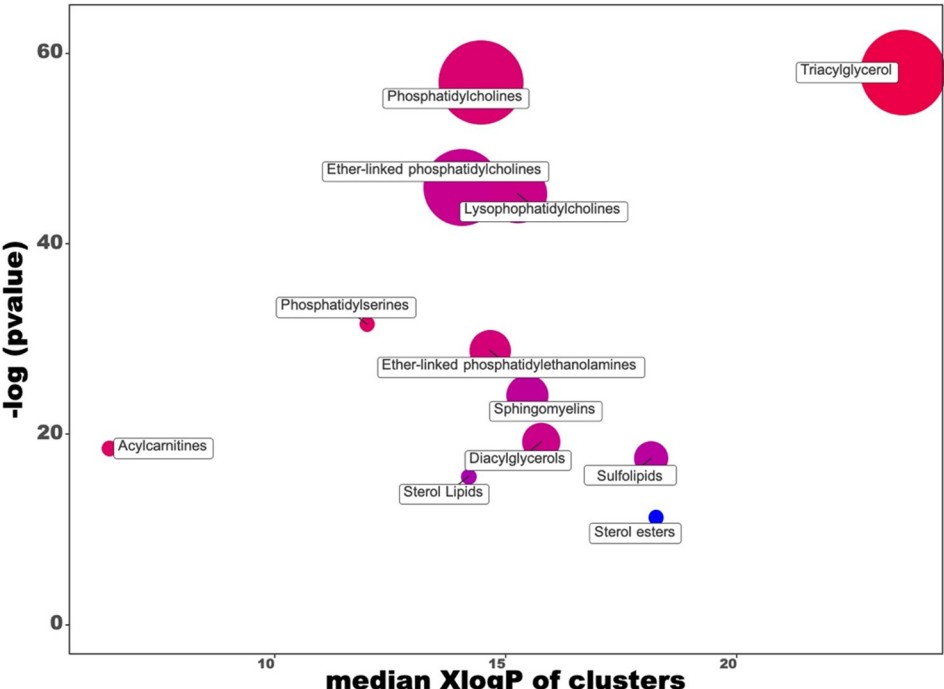

**Fig 2. ChemRICH set enrichment statistics plot built considering the compounds significantly discriminating the comparison L+P *vs* CTRL diets.** Each node reflects a significantly altered cluster of metabolites. Enrichment *p*-values are given by the Kolmogorov–Smirnov test. Node sizes represent the total number of metabolites in each cluster set. The node color scale shows the proportion of increased (red) or decreased (blue) compounds in L+P-diet compared to CTRL. Purple-color nodes have both increased and decreased metabolites.

phosphatidylcholines (30 compounds), ether-linked phosphatidylcholines (25 compounds), and lysophosphatidylcholines (14 compounds). Interestingly, almost 69% of the discriminant compounds were found to be up-accumulated in the L+P diet group, thus revealing a clear impact of the feeding system on the lipidomic profile of pig *Longissimus thoracis* muscle.

As a subsequent evaluation, a PCA score plot built considering the discriminant compounds was inspected (S5 Fig), showing a good degree of separation between the two groups, with the two principal components able to explain the 53.6% of the cumulative variability. Finally, a chemical similarity enrichment (ChemRICH) was used as an alternative to biochemical pathway mapping when considering the previously reported discriminant compounds. ChemRICH is based on structure similarity and chemical ontologies to map all known metabolites and name metabolic modules. The ChemRICH set enrichment statistics plot is reported in Fig 2, while Table 1 reports the main cluster sizes and the key compounds. As can be observed from this figure, the sterol esters lipid cluster showed an absolute decreasing trend, although with a low cluster size (i.e., 3 discriminant compounds). Regarding the other lipid clusters, those presenting the largest sizes were glycerophospholipids (including phosphatidylcholines, ether-linked phosphatidylcholines, and lysophosphatidylcholines) and triacylglycerols (32 compounds). Interestingly, these subclasses showed both increased and decreased significant metabolites (Table 1).

In our experimental conditions, the L+P diet was enriched with extruded linseed and plant extracts from grape-skin and oregano as polyphenols sources. The inclusion of linseeds in the pig diet is reported to improve meat quality by strengthening the content of several compounds, such as α-linolenic, eicosatrienoic, eicosapentaenoic, and docosapentaenoic fatty

**Table 1. Results of the ChemRICH enrichment analysis for the altered metabolites following the comparison L+P *vs* CTRL diets.**

| Cluster name | Cluster size | *p*-values (FDR) | Key compound | Altered metabolites | Increased | Decreased |
|---|---|---|---|---|---|---|
| Triacylglycerol | 32 | 7.6E-25 | TG_8:0_8:0_34:6 | 32 | 25 | 7 |
| Phosphatidylcholines | 31 | 1.1E-24 | PC 32:3 | 31 | 19 | 12 |
| Ether-linked phosphatidylcholines | 25 | 4.4E-20 | PC O-31:6 | 25 | 12 | 13 |
| Lysophophatidylcholines | 14 | 6.6E-20 | LPC 36:6 | 14 | 7 | 7 |
| Phosphatidylserines | 3 | 4.6E-14 | PS 38:6 | 3 | 2 | 1 |
| Ether-linked phosphatidylethanolamines | 7 | 6.1E-13 | PE O-16:3_22:6 | 7 | 4 | 3 |
| Sphingomyelins | 7 | 5.9E-11 | SM 41:4;2O | 7 | 3 | 4 |
| Diacylglycerols | 6 | 6.7E-09 | DG 13:0_44:9 | 6 | 3 | 3 |
| Acylcarnitines | 3 | 1.2E-08 | CAR 14:3 | 3 | 2 | 1 |
| Sulfolipids | 5 | 3.0E-08 | SL 21:3;O/26:2;O | 5 | 2 | 3 |
| Sterol Lipids | 3 | 1.9E-07 | ST 24:1; O4/15:0 | 3 | 1 | 2 |
| Sterol esters | 3 | 1.20E-05 | 16:1 Cholesterol ester | 3 | 0 | 3 |

acids, with no significant impact on the growth performance of pigs [30]. Our findings revealed a great representation of phosphatidylcholines and triglycerides in the ChemRICH plot. Accordingly, although triglycerides represent the major lipid fractions in meat, also phospholipids can contribute to lipid oxidation phenomena. A reasonable explanation is that phospholipids are characterized by a higher amount of polyunsaturated fatty acids than triglycerides [31, 32]. Also, previous studies on pork and beef fresh meat showed that total PUFA in triglycerides represent 4.5–14%, while this fraction reaches 37–47% in the polar lipid fraction (phospholipids) [32]. Therefore, phospholipids can be considered strong contributors to the development of lipid oxidation and rancidity of the product, potentially affecting its quality attributes. Our findings showed an overall increase of triglycerides (increased ratio: 0.8), with a corresponding rise in phosphatidylcholines (increased ratio: 0.6), thus potentially promoting the oxidation of meat lipids as a consequence of the L+P diet. This was also confirmed by inspecting other discriminant compounds characterizing the lipidomic dataset. In this regard, we found a substantial up accumulation of fatty acid 20:5 (Log2FC = 2.7), whilst ether-linked phosphatidylcholines and lysophosphatidylcholines showed both an increased ratio of 0.5. Although several controversies still exist in the academic community about meat oxidation and lipid composition, the unsaturation of fat remains one of the most important parameters related to the oxidative susceptibility of meat [32]. Another intriguing finding was obtained when considering the down-accumulation of some cholesterol esters (Fig 2), with 16:1 cholesterol ester classified as a key compound in the ChemRICH plot. Cholesterol is an essential component of animal tissue, highly prone to oxidation processes. In particular, the process of cholesterol oxidation is similar to those involving other unsaturated lipids, considering that it is based on the reaction with reactive oxygen species (ROS) [32]. Therefore, taken together, our findings seem to suggest a complex interplay of different metabolic routes involving lipids and following the L+P diet, with glycerophospholipids and glycerolipids being the most affected classes of compounds.

## Discrimination between CTRL and L diets based on lipidomic profiling

Afterward, the same statistical workflow was used to compare the diet enriched with extruded linseed (5% of feed) (L) with the control diet (CTRL). Overall, the combination of both univariate and multivariate statistical approaches allowed identifying 152 discriminant compounds. In this regard, a comprehensive table reporting Log2FC values, *p*-values, VIP scores, together with annotation information (such as formula, adduct, and annotation level), can be found in

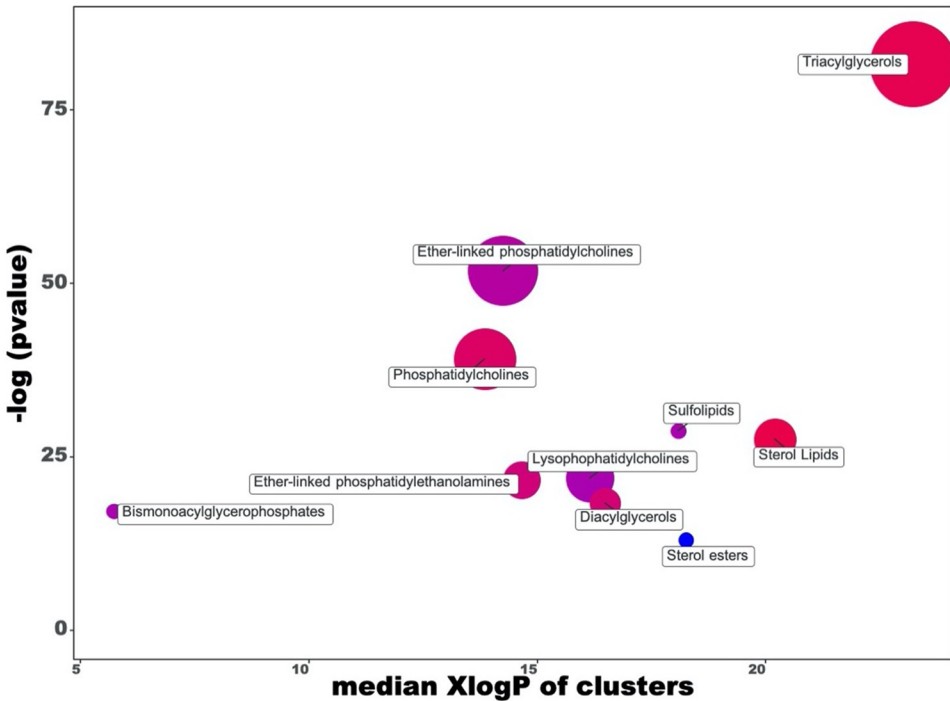

**Fig 3. ChemRICH set enrichment statistics plot considering the comparison L *vs* CTRL diets.** Each node reflects a significantly altered cluster of metabolites. Enrichment *p*-values are given by the Kolmogorov–Smirnov test. Node sizes represent the total number of metabolites in each cluster set. The node color scale shows the proportion of increased (red) or decreased (blue) compounds in L-diet compared to CTRL. Purple-color nodes have both increased and decreased metabolites.

S2 Table. Overall, triacylglycerols and ether-linked phosphatidylcholines were the most represented among the discriminant compounds, recording 45 and 28 annotations, respectively. Also, 66.4% of the compounds detected were up-accumulated in the L-diet group, thus revealing a significant impact of the linseed supplementation on the lipidomic profile of pig *Longissimus thoracis* muscle. Thereafter, a ChemRICH plot was used to group the up- and down-accumulated compounds in specific biochemical classes. The output of the ChemRICH plot analysis is reported in Fig 3, while the corresponding PCA score plot (explaining 43% of the total variability with two principal components) can be observed in S6 Fig. As can be observed from this figure, few differences were found in the metabolic nodes affected by L compared to the L+P diet; in fact, triglycerides (n = 45) and ether-linked phosphatidylcholines (n = 28) were again the most represented cluster of compounds. In this regard, different increasing/decreasing trends were observed, with triglycerides being characterized by an increased ratio of 0.8, while ether-linked phosphatidylcholines showed an overall decreasing trend (increased ratio: 0.4) (Table 2).

Interestingly, we confirmed an absolute down-accumulation of cholesterol esters (*p*-value = 2.3 x 10$^{-6}$); therefore, it is possible to attribute this effect to the L diet rather than the L +P diet. Also, several common key compounds were detected when comparing the Chem-RICH plot findings of L+P *vs*. CTRL and L *vs*. CTRL comparisons (Tables 1 and 2). Therefore, the statistical analyses on the comparison L *vs*. CTRL and L+P *vs*. CTRL diets allowed to observe few global differences; however, as a general consideration, it is possible to notice the involvement of triglycerides following supplementation with linseeds, whilst the combined

**Table 2. Results of the ChemRICH enrichment analysis for the altered metabolites following the comparison L *vs* CTRL diets.**

| Cluster name | Cluster size | *p*-values (FDR) | Key compound | Altered metabolites | Increased | Decreased |
|---|---|---|---|---|---|---|
| Triacylglycerols | 45 | 3.40E-35 | TG_8:0_8:0_34:5 | 45 | 34 | 11 |
| Ether-linked phosphatidylcholines | 28 | 1.50E-22 | PC O-31:6 | 28 | 11 | 17 |
| Phosphatidylcholines | 21 | 3.50E-17 | PC 32:3 | 21 | 14 | 7 |
| Sulfolipids | 3 | 8.00E-13 | SL_21:3;O/26:2;O | 3 | 1 | 2 |
| Sterol Lipids | 9 | 2.20E-12 | ST_24:1;O4/17:1 | 9 | 7 | 2 |
| Lysophosphatidylcholines | 12 | 4.80E-10 | LPC 36:6 | 12 | 4 | 8 |
| Ether-linked phosphatidylethanolamines | 7 | 5.50E-10 | PE O-9:0_28:6 | 7 | 4 | 3 |
| Diacylglycerols | 5 | 1.40E-08 | DG 13:0_44:9 | 5 | 3 | 2 |
| Bismonoacylglycerophosphates | 3 | 3.90E-08 | BMP 2:0_17:0 | 3 | 1 | 2 |
| Sterol esters | 3 | 2.30E-06 | 14:1 Cholesterol ester | 3 | 0 | 3 |

addition of the polyphenol-extracts and linseeds produced a major alteration of different glycerophospholipid subclasses.

## Exclusive up-accumulated and discriminant compounds of L and L+P vs CTRL

Considering the few differences emerging from the lipidomic mapping approach, a Venn diagram was used to plot those UP-accumulated compounds in common or exclusive of each diet (i.e., L and L+P). The Venn diagram is reported in Fig 4. As can be observed from this figure, 55.6% of significantly UP-accumulated compounds (74 lipid subclasses) were found to be in common between L and L+P diets. Interestingly, each experimental diet determined a selective UP accumulation of specific compounds. In fact, the L+P diet provided, among the others, 15 glycerophospholipids (mainly glycerophosphocholines) and 8 sphingolipids, whilst the L diet determined a marked and exclusive UP accumulation of glycerolipids (18 triacylglycerols).

Besides, an additional multivariate statistical model based on the comparison "L vs L+P" was carried out in order to exclusively evaluate the impact of phenolic extracts (characterizing the L+P experimental diet) on the changes of lipidomic profiles in meat. Overall, as clearly shown in Fig 1 and as confirmed by inspecting the ChemRICH plot for the comparisons against the CTRL diet, the untargeted lipidomic profile of meat samples belonging to L and L+P diets was very similar, showing a certain degree of overlapping. This latter point was also confirmed by inspecting the PCA score plot for the comparison L vs L+P diet (not shown), demonstrating a low ability of discrimination of meat samples belonging to L and L+P experimental groups. Therefore, in order to discern among the best discriminant markers of the lipidomic differences potentially induced by the addition of polyphenols, a sparse PLS-DA

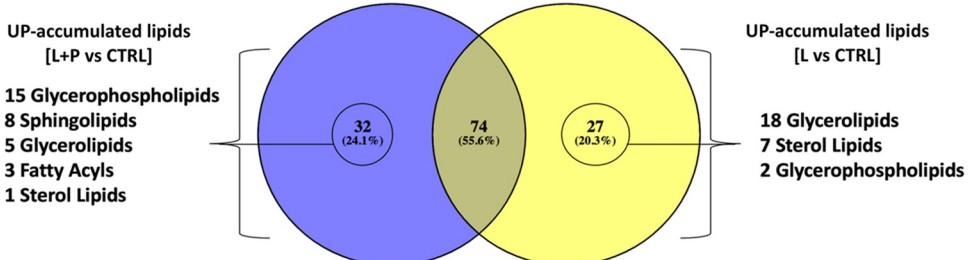

**Fig 4. Venn diagram considering the significant (p < 0.05) and UP-accumulated lipid subclasses following both L +P *vs* CTRL (purple color) and L *vs* CTRL (yellow color) pairwise comparisons.**

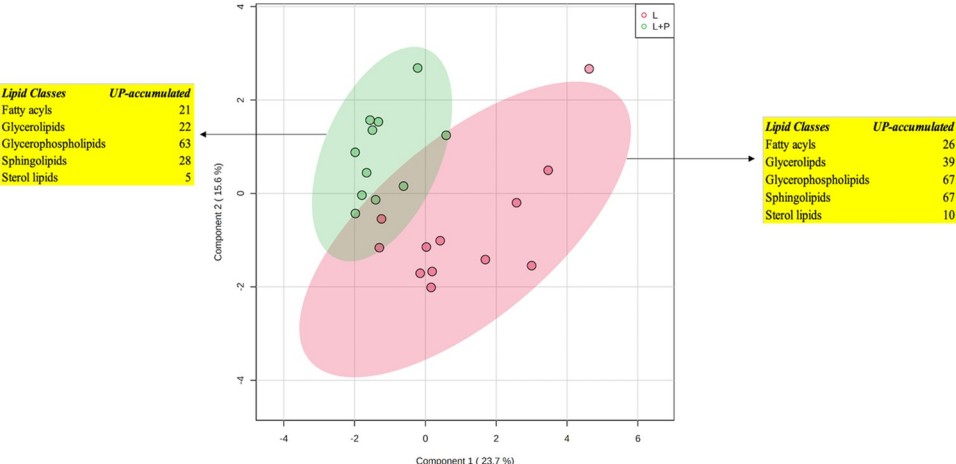

**Fig 5. sPLS-DA score plot considering the comparison "L" vs "L+P" diets.** The discriminant and up-accumulated lipid classes for each group are also provided.

(sPLS-DA) algorithm was used to reduce the number of variables (i.e., discriminant lipids) and to produce a robust and easy-to-interpret model. The sPLS-DA score plot for the comparison L vs L+P is provided as Fig 5.

As can be observed from the figure, a not complete separation was revealed between the different samples, thus confirming a quite similar lipidomic profile and a low degree of discrimination as provided by the addition of polyphenols. Finally, the most discriminant variables driving the separation between L and L+P groups were selected according to the VIP approach. Overall, 351 compounds showed (S2 Table) VIP score > 1 (i.e., extremely discriminant in the prediction model); in particular, 59.5% of the discriminant lipids was up-accumulated in the "L" group (namely 26 fatty acyls, 39 glycerolipids, 67 glycerophospholipids, 67 sphingolipids, and 10 sterol lipids), whilst the remaining 40.5% of the compounds exclusively characterized the "L+P" group (namely 21 fatty acyls, 22 glycerolipids, 63 glycerophospholipids, 28 sphingolipids, 5 sterol lipids and 3 other compounds). Overall, the most up-accumulated compound characterizing the "L" group was a glycerolipid, namely DG O-18:0_28:7 (VIP score: 4.52; Log2FC: 16.39), whilst the addition of polyphenols (L+P) determined a marked up-accumulation of the sphingolipid SL 10:1;O/18:2;O (VIP score 1.86; Log2FC: 15.16) followed by the glycerophospholipid PC O-50:0 (VIP score: 2.76; Log2FC: 14.32). It was interesting to notice that L and L+P experimental diets differed mainly in the accumulation of glycerolipids (39 vs 22, respectively) and sphingolipids (67 vs 28, respectively). The lower number of up-accumulated sphingolipids outlined for the L+P group might be related to the potential inhibition of phenolic compounds characterizing oregano and grape skin extracts towards the enzymes involved in their de novo synthesis [33], although further works (including both detailed phenolic profiling of the experimental diets and in silico approaches) are mandatory to confirm this hypothesis.

Taken together, our findings demonstrated an impact of each experimental diet not only on the nutritional quality of meat but also a possible modulation of gene expression. Accordingly, some previous works showed that dietary supplementation with *n-3* PUFA can influence the expression of genes involved in the inflammatory response, oxidation, muscle development and differentiation, muscle protein metabolism, glucose metabolism, and fatty acid biosynthesis [4, 12]. Besides, few works existing in the literature have evaluated the impact of polyphenols added to the diet on the lipidomic profile of the meat. Overall, the most important studies

in this research area correlated polyphenols to the expression of genes involved in lipid metabolism, inflammation, and extracellular matrix remodeling [34, 35]. Generally, phospholipid composition is strictly related to membrane biophysical properties. Therefore, it is possible to postulate that changes in the incorporation of polyunsaturated acyl chains into phospholipids might affect lipid transport across cellular membranes. For example, a previous work [36] showed that challenging mammalian membrane homeostasis by dietary lipids led to a robust lipidomic remodeling, thus preserving the membrane physical properties. Starting from these considerations, our results seem to sustain a mechanism leading to membrane homeostasis based on a lipidome change in response to dietary lipids and aimed at preserving the functional membrane phenotypes. Indeed, the physiology of biological membranes requires the maintenance of specific physicochemical properties, which must be buffered from external perturbations. As reported by Leventhal et al. [36], exogenous PUFAs are rapidly and extensively incorporated into membrane lipids, thus determining a reduction in membrane packing. However, these effects are fastly compensated both in culture and *in vivo* by a wide lipidome remodeling, mostly involving up-accumulation of saturated lipids and cholesterol derivatives, to compensate for the fluidizing effect of PUFA-containing species and thus recovering the correct membrane packing [36]. This lipidomic compensation is mostly mediated by the sterol regulatory machinery, whose pharmacological or genetic removal determines a decreased cellular fitness when membrane homeostasis is challenged by dietary lipids. In this work, we found an exclusive up-accumulation of seven sterol lipids, namely five acylhexosyl campesterol and two acylhexosyl cholesterol derivatives, as resulted from L experimental diet. In this regard, the five campesterol derivatives observed are likely derived from linseeds, as reported in the scientific literature [37]. Therefore, although in mammalian membranes cholesterol is the predominant sterol, we hypothesized that phytosterols characterizing the functional diet L were used to counteract the accumulation of phospholipids (derived from *n*-3 PUFA-rich diet) in the cellular membrane. A previous work [38] showed that incorporating the phytosterols into the phospholipid monolayers was able to increase their condensation, with phytosterols presenting a higher affinity towards phosphatidylcholines compared to phosphatidylethanolamines. Also, these authors reported that phytosterols interacted more strongly with phospholipids possessing longer and saturated chains. Therefore, untargeted lipidomics seems to suggest selected changes of the lipid composition, involving a complex interplay between glycerolipids, phospholipids, and sterol lipids (both cholesterol and plant sterols from diet), and likely based on the homeostatic remodeling of the cellular membrane. However, to support this latter hypothesis, additional and ad-hoc works based on the consideration of more time-points are needed, to confirm these changes over time.

## Correlating lipidomics and transcriptomics results

In a previous work [12], we identified a total of 157 DEGs in the comparison L *vs*. CTRL, of which 6 were up-regulated and 151 down-regulated in L compared to CTRL. The complete list of DEGs in L *vs*. CTRL and their expression level are reported in S3 Table. The cluster analysis performed for the DEGs and compounds found in L *vs*. CTRL highlights two dimensions explaining 46% of the total variance: 30% of the total variance is explained by Dimension 1 (Dim1), and 16% by Dim2. The complete list of dimensions and their distance values are reported in S4 Table. When plotting the results of the cluster analysis (Fig 6), pigs fed L and CTRL are separated.

The results of the unsupervised PCA analysis for the comparison L *vs*. CTRL and the correlation of the individual eigenvalues with DEGs and lipids are reported in S5 Table. The final correlation between genes and lipids evidenced 14 genes and 36 compounds as the most

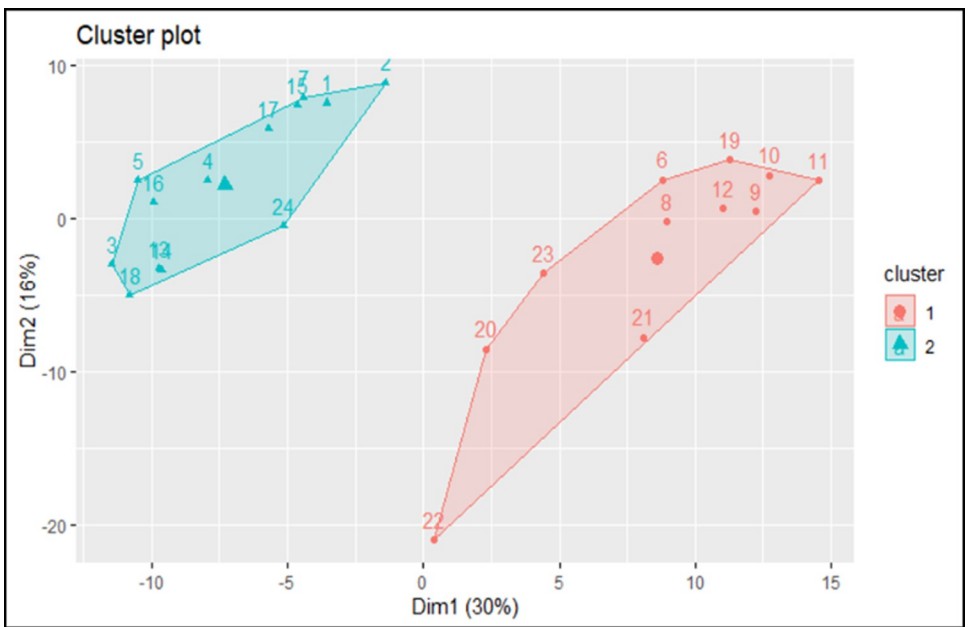

**Fig 6. Cluster analysis of L vs CTRL comparison.** Numbers represent the pigs sampled and different colours indicate the clusters. Cluster 1 = CTRL; Cluster 2 = L.

significantly correlated and discriminating the two diets (S6 Table). The 14 DEGs were only present in this comparison, while among the lipids only 4 are exclusive in this comparison and 32 have also been found between L+P *vs*. CTRL. Some genes correlate with more than one lipid for a total of 85 significant correlations, of which 65 were negative and 20 positive. Those genes showing an *r* value > 0.6 or < -0.6 and having a significant *p* are listed in Table 3. The lipid classes establishing a high number of correlations were predominantly

**Table 3. List of genes showing correlation with lipids, in the two comparisons.**

| Symbol | Gene name | L *vs* CTRL | Correlation with lipids (n) | | |
|---|---|---|---|---|---|
| | | | total | positive | negative |
| BCL2L13 | BCL2-Like 13 (Apoptosis Facilitator) | ↓ | 1 | 0 | 1 |
| BTBD6 | Glucocorticoid Receptor AF-1 Coactivator-1 | ↓ | 1 | 0 | 1 |
| DNAJB1 | DnaJ Heat Shock Protein Family (Hsp40) Member B1 | ↑ | 13 | 13 | 0 |
| DUSP26 | Dual Specificity Phosphatase 26 | ↓ | 1 | 0 | 1 |
| HIF1AN | Hypoxia Inducible Factor 1 Subunit Alpha Inhibitor | ↓ | 27 | 0 | 27 |
| KCNJ11 | Potassium Inwardly Rectifying Channel Subfamily J Member 11 | ↓ | 11 | 0 | 11 |
| KCNJ12 | Potassium Inwardly Rectifying Channel Subfamily J Member 12 | ↓ | 9 | 0 | 9 |
| LOC100152722 | - | ↓ | 1 | 0 | 1 |
| MBD3 | Methyl-CpG Binding Domain Protein 3 | ↓ | 7 | 0 | 7 |
| PCBP2 | Poly(rC) Binding Protein 2 | ↓ | 2 | 0 | 2 |
| TMSB4X | Thymosin Beta 4 X-Linked | ↑ | 6 | 6 | 0 |
| TUBGCP2 | Tubulin Gamma Complex Associated Protein 2 | ↓ | 1 | 0 | 1 |
| VLDLR | Very Low Density Lipoprotein Receptor | ↓ | 1 | 0 | 1 |
| ZBTB47 | Zinc Finger And BTB Domain Containing 47 | ↓ | 4 | 0 | 4 |

CTRL = control diet; L = diet supplemented with linseed. ↑ = genes up-regulated in L as compared to CTRL; ↓ = genes down-regulated in L as compared to CTRL.

glycerolphospholipids (78 correlations), followed by N-acyl amino acids (3), sphingolipids (2); fatty acyls (1), and phosphatidylethanolamines (1) (S6 Table). Among the significant correlations between genes and lipids, a negative relationship exists between an increased accumulation of glycerophospholipids and a decreased expression of two members of the Potassium Inwardly Rectifying Channel Subfamily J (i.e., *Potassium Inwardly Rectifying Channel Subfamily J Member 11- KCNJ11*, and *Potassium Inwardly Rectifying Channel Subfamily J Member 12- KCNJ12*) in L as compared to CTRL. Also, among glycerophospholipids, some glycerophosphoethanolamines showed a negative correlation with *KCNJ11* and *KCNJ12* gene expression, with *r* values ranging from -0.60 to -0.72 ($p < 0.01$). These correlations agree with the literature, where altered membrane lipids and biophysical properties influence membrane ion channel gene transcription and function [39, 40]. In particular, those alterations in the membrane lipid composition are associated with changes in the transcription levels of some potassium channels (also comprising *KCNJ11*) in the heart of rats used as models for diabetes-related perturbances [39]. These perturbations in ion channel expressions also alter calcium channels, causing potential changes in membrane receptors controlled by those ions. This cascade may help to explain the downregulation in L of the *Very Low Density Lipoprotein Receptor* (*VLDLR*) gene, which codes for a receptor mediating the endocytosis of VLDLs and responding to the concentrations of free calcium ions [41]. The mRNA level of the *VLDLR* gene negatively correlates ($r$ = -0.604; $p$ = 0.0018) with the amount of a compound pertaining to ether-linked phosphatidylcholine, which is more accumulated in pigs fed L. *VLDLR* gene is poorly expressed in preadipocytes while it is highly expressed in mature adipocytes [42]. These results suggest that *VLDLR* may have an important role in adipogenesis, and the literature indicates that a deficiency in *VLDLR* expression may prevent obesity onset [43]. Furthermore, *VLDLR* deficiency is also observed to reduce inflammation and endoplasmic reticulum stress in adipose tissue, together with a decrease in macrophage infiltration, especially macrophages expressing pro-inflammatory markers [44]. The downregulation of the *VLDLR* gene in L samples and its negative correlation with the increased accumulation of an ether-linked phosphatidylcholine compound in L may indicate that this diet inhibits adipocyte maturation in muscle tissue, protecting against fat-related inflammation processes. In agreement with those hypotheses, L also downregulated the expression of other genes stimulating inflammation, such as *Hypoxia Inducible Factor 1 Subunit Alpha Inhibitor* (*HIF1AN*) [45], and adipogenesis, as *BCL2 Like 13* (*BCL2L13*) [46]. Based on these observations, the gene-compound correlations seem to strengthen the hypothesis already reported in Sirri et al. [12] that L may prevent adipogenesis and the related inflammation processes, decreasing the expression of genes related to adipogenesis and inflammation.

The comparison L+P *vs*. CTRL evidenced 93 DEGs, 73 of which were up-regulated and 20 down-regulated in L+P. The complete list of genes and levels of expression can be found in S7 Table. The cluster analysis showed in L+P *vs*. CTRL that the pigs fed the two diets significantly differed, as in Fig 7. The analysis reported that 59% of the variance is explained in the two first dimensions (dim1 41.3%; dim 2 17.7% of variance). The full results are in S8 Table. The result of the unsupervised PCA analysis for the comparison L+P *vs*. CTRL and the correlation of the individual eigenvalues with DEGs and lipids are reported in S9 Table. In L+P *vs*. CTRL, the final correlations between genes and lipids evidenced 17 genes and 66 lipids as the most significantly correlated and discriminating the two diets (S10 Table). The DEGs were only present in this comparison, while among the lipids, 16 are exclusive in this comparison, and 50 have also been found between L *vs*. CTRL. Some genes correlated with more than one lipid for a total of 165 correlation, 155 with a positive *r* and 12 having negative *r* coefficients. Those genes showing an *r* > 0.6 or < -0.6 and having a significant *p* are listed in Table 4.

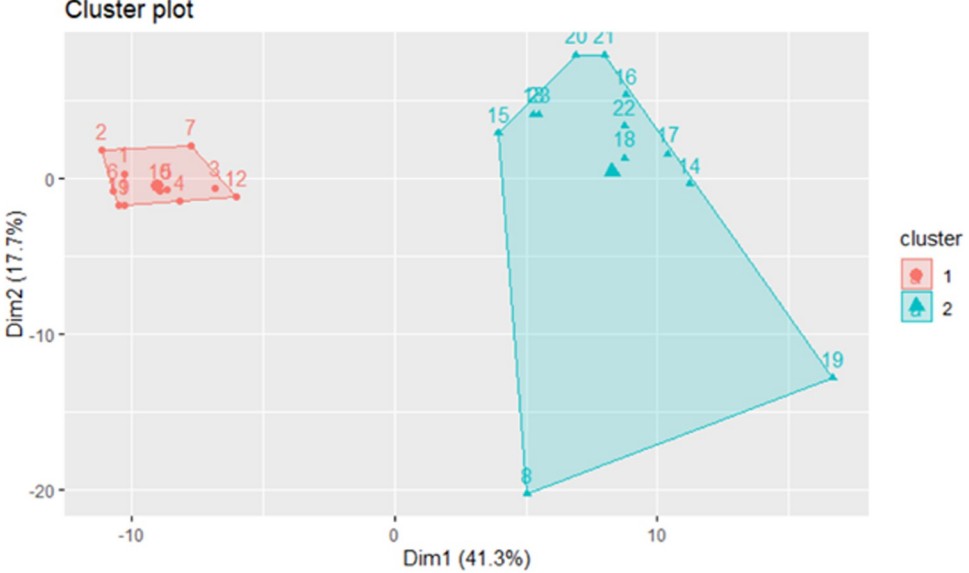

**Fig 7. Cluster analysis of L+P vs CTRL comparison.** Numbers represent the pig sampled and different colours indicate the clusters. Cluster 1 = CTRL; Cluster 2 = L+P.

The lipids establishing a high number of correlations were predominantly glycerolphospholipids (125 correlations), followed by glycerolipids (11), bile acids (11); sphingolipids (9); sulfonolipids (7); amino acids and derivatives (2), and sterol lipids (1). Among the genes found up-regulated in L+P, those more specifically involved in lipid metabolism and glucose homeostasis were *Apolipoprotein E* (*APOE*), *Caveolae Associated Protein 1* (*CAVIN1*), *Coiled-Coil*

**Table 4. List of genes showing significant correlation with lipids.**

| Symbol | Gene name | L+P *vs* CTRL | Correlation with lipids (n) | | |
|---|---|---|---|---|---|
| | | | total | positive | negative |
| APOE | Apiloprotein E | ↑ | 7 | 7 | 0 |
| APP | Amyloid Beta Precursor Protein | ↑ | 10 | 10 | 0 |
| CAVIN1 | Caveolae Associated Protein 1 | ↑ | 6 | 5 | 1 |
| CCDC80 | Coiled-Coil Domain Containing 80 | ↑ | 7 | 7 | 0 |
| CDH13 | Cadherin 13 | ↑ | 13 | 12 | 1 |
| COL14A1 | Collagen Type XIV Alpha 1 Chain | ↑ | 1 | 1 | 0 |
| COL1A2 | Collagen Type I Alpha 2 Chain | ↑ | 6 | 6 | 0 |
| COL3A1 | Collagen Type III Alpha 1 Chain | ↑ | 3 | 3 | 0 |
| ELN | Elastin | ↑ | 17 | 17 | 0 |
| ELOVL5 | Fatty Acid Elongase 5 | ↑ | 5 | 5 | 0 |
| LOC100624077 | - | ↑ | 1 | 1 | 0 |
| MMP2 | Matrix Metallopeptidase 2 | ↑ | 18 | 18 | 0 |
| NES | Nestin | ↑ | 2 | 2 | 0 |
| SCD | Stearoyl-CoA Desaturase | ↑ | 18 | 18 | 0 |
| SMYD1 | SET And MYND Domain Containing 1 | ↑ | 34 | 25 | 9 |
| SPARC | Secreted Protein Acidic And Cysteine Rich | ↑ | 6 | 6 | 0 |
| TIMP2 | Metallopeptidase Inhibitor 2 | ↑ | 14 | 14 | 0 |

CTRL = control diet; L+P = diet supplemented with linseed and plant extracts. ↑ = genes up-regulated in L+P as compared to CTRL; ↓ = genes down-regulated in L+P as compared to CTRL.

*Domain Containing 80* (*CCDC80*), *Stearoyl-CoA Desaturase* (*SCD*), and *ELOVL Fatty Acid Elongase 5* (*ELOVL5*). Recently, a multi-omics analysis in mice elucidated the role of *CCDC80* in regulating arachidonic acid and α-linolenic acid metabolism [47]. At the same time, elongases and desaturases (*SCD* and *ELOVL5)* are essential players in fatty acid elongation and desaturation, also regulating fatty acid functions and metabolic fates [48]. An increase in *ELOVL5* and *SCD* in the present study has been correlated with a higher accumulation of diacylglycerol (DG) and phosphatidylcholine (PC) (S10 Table). Despite being the first time that these associations have been studied in pigs, previous studies in mice and humans have been reported that lipid-dependent signalling might influence and nuclear diacylglycerol (DG) pathway in the cell nucleus, stimulating DG from PC, as a second messenger in the nucleus [49]. According to Goto et al. [50], at least 50 structurally distinct molecular species are included among DGs, whose fatty-acyl groups can be polyunsaturated, di-unsaturated, mono-unsaturated, or saturated. According to the same review, the different DG species can be involved in different pathways in the regulation of cell metabolism, growth and apoptosis; however full knowledge of how DG species regulate cell metabolism, growth and apoptosis is still lacking [50]. In this context, it seems that L+P stimulated the higher expression of genes involved in lipid biosynthesis, according to the results of Sirri et al. [12], and those involved in long-chain PUFA (*APOE*, *CAVIN1*, *CCDC80*, *SCD*, and *ELOVL5*), might have regulated DG and PC compounds accumulation in the cell, increasing cells activity and signalling. The high expression of *NES* seems to confirm the high nuclear activity since NES protein has been found to mediate signal-dependent transport of diacylglycerol kinases (DGK) from the nucleus back into the cytoplasm [50].

Moreover, other genes up-regulated in L+P as compared to CTRL and correlated to the lipid classes are known to be linked to the formation, function, and remodeling of the extracellular matrix (ECM), such as *Cadherin 13* (*CDH13*), *Collagen Type XIV Alpha 1 Chain* (*COL14A1*), *Collagen Type I Alpha 2 Chain* (*COL1A2*), *Collagen Type I Alpha 2 Chain* (*COL1A1*), *Collagen Type III Alpha 1 Chain* (*COL3A1*), *Elastin* (*ELN*), *Matrix Metallopeptidase 2* (*MMP2*), *TIMP Metallopeptidase Inhibitor 2* (*TIMP2*), *Secreted Protein Acidic And Cysteine Rich* (*SPARC*). It is currently known that the increase of adipose tissue is dependent on adipogenesis, angiogenesis and extracellular matrix remodeling [51]. However, the structural organization of the matrix remains poorly understood, especially for the adipogenic ECM [52]. According to the same authors, the adipogenic differentiation process comprises the commitment and differentiation of mesenchymal stem cells (MSC) into the adipogenic lineage. During these steps, adipocytes excrete the proteins taking part in the ECM, together with several other molecular factors and signaling cascades known to play a critical role during adipogenic differentiation and ECM development [52]. In a study by Mariman and Wang [53], conducted on MSC-derived adipogenic ECM, several genes have been related to adipogenic differentiation and ECM development, confirming an association between the expression of these DEGs and an effect on the cellular structure and integrity. Specifically, *COL14A1*, *COL1A1*, *TIMP2* pertained to adipogenic ECM formation and maintenance, and are the same found up-regulated in L+P in our study, thus supporting the role of the diet enriched in extruded linseed and plant extracts (L+P) in promoting adipogenic ECM formation and functioning. *CDH13* and *MMP2* have been found in studies on murine models to be importantly involved in the differentiation potential of adipocytes and able to affect lipid metabolism during adipogenesis, therefore considered markers for plasticity and health status of the adipose tissue [51, 54]. Consequently, it is not surprising to have found in our study a robust positive correlation between genes involved in adipose tissue differentiation and adipogenic ECM function and the lipid classes glycerolphospholipids, glycerolipids, sphingolipids, and sulfonolipids, known to be involved in cell membrane maintenance and protection [55]. In a recent study by Xu

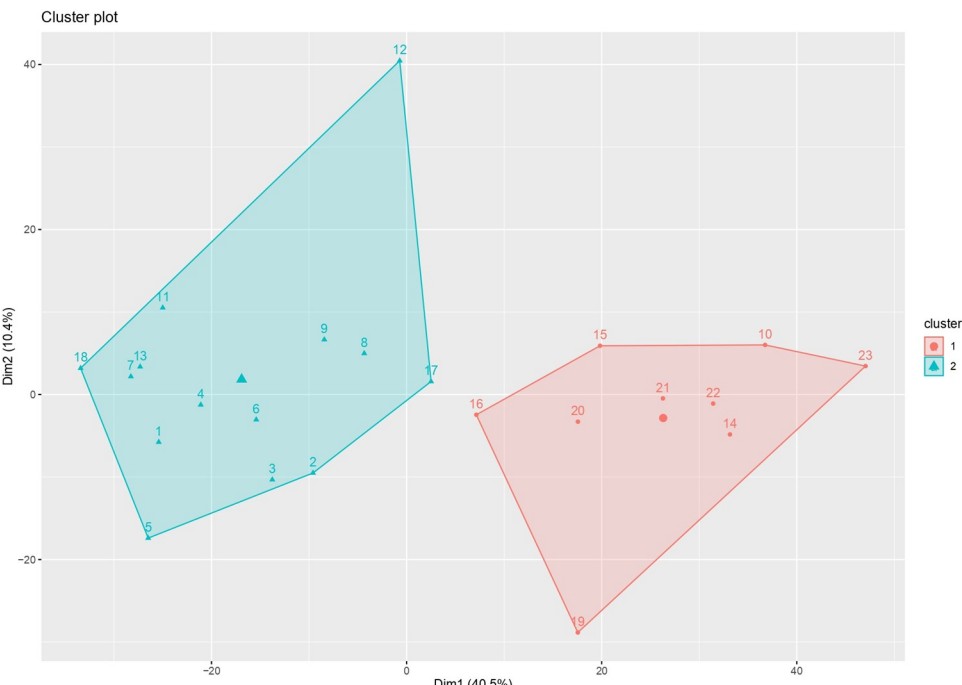

**Fig 8. Cluster analysis of L vs L+P comparison.** Numbers represent the pigs sampled and different colours indicate the clusters. Cluster 1 = L+P; Cluster 2 = L.

et al. [56] in mice with glycerol-induced intramuscular fat infiltration, the differentially expressed genes were those involved in the metabolism of glycerolipids, glycerophospholipids, and sphingolipids, in parallel with the accumulation in the fat-infiltrated muscle of the same lipid classes glycerolipids, glycerophospholipids, fatty acyls, and sphingolipids.

Finally, an integrated lipidomics and transcriptomic analysis was also performed for L *vs.* L +P. In our previous study [4], a total of 1102 DEGs were evidenced for this comparison, 628 of which were up-regulated and 474 down-regulated in L. The complete list of genes and levels of expression can be found in S11 Table. The cluster analysis showed in L. *vs.* L+P that the pigs fed the two diets significantly differed, as in Fig 8. The analysis reported that 50.9% of the variance is explained in the two first dimensions (dim1 40.5%; dim 2 10.4% of variance). The full results are in S12 Table.

The result of the unsupervised PCA analysis for the comparison L *vs.* L+P and the correlation of the individual eigenvalues with DEGs and lipids are reported in S13 Table. The results of the PCA are in agreement with the lack of complete separation revealed with PLS-DA for L *vs.* L+P. Indeed, among the lipids and DEGs considered in the PCA, none of the lipids showed to be important for discriminating the two diets, and only 1 DEG correlated with PC1 eigenvalues with an $r > |0.60|$. For this reason, no correlation analysis was performed between the lipids and the genes. This result, therefore, indicates that L *vs.* L+P diets comparison did not lead to large differences in the lipidomic profile of the samples, and most of the differences observed in the transcriptome are related to genes that are not involved in lipid metabolism while L+P activated several genes involved in muscle development and physiology [4].

## Conclusions

In this study, we demonstrated that a comprehensive and combined lipidomics/transcriptomics approach could be a great tool to evaluate the impact of functional diets, namely one

supplemented with *n-3* PUFA from extruded linseed (L) and another one based also on pheno-lic compounds from grape skin extracts (L+P), on the functional quality of pig *Longissimus thoracis* muscle. In particular, the untargeted UHPLC-QTOF mass spectrometry approach allowed us to observe changes of several lipid classes, such as glycerolipids, phospholipids, and sterol lipids (both cholesterol and plant sterols from diet). Looking at the comparisons with the CTRL diet, the L+P diet significantly increased 23 lipids (i.e., 15 glycerophospholipids and 8 sphingolipids), while the L diet was associated to a strong up-accumulation of glycerolipids. Finally, our findings showed potential correlations between the lipidomic profile and the modulation of gene expression, with the L diet associated to the prevention of both adipogenesis and inflammation processes, and the L+P diet promoting lipids' biosynthesis and adipogenic extracellular matrix functionality. Further research, based on ad-hoc and targeted mass spectrometry studies is warranted to select a ristricted pool of marker compounds directly correlated to each functional diet.

## Supporting information

**S1 Fig.** Ultra-high-performance liquid chromatography coupled to high resolution mass spectrometry base peak chromatograms of meat sample extracts obtained using positive (A) and negative (B) ionization modes.
(TIF)

**S2 Fig.** Scatter score plots for PCA models built from positive (A) and negative (B) ionization modes.
(TIF)

**S3 Fig.** Scatter score plots for PCA models built from positive (A) and negative (B) ionization modes.
(TIF)

**S4 Fig.** Scatter score plots for PCA models built from positive (A) and negative (B) ionization modes.
(TIF)

**S5 Fig. PCA score plot built considering the discriminant compounds of the comparison L +P vs CTRL diets.**
(TIF)

**S6 Fig. PCA score plot built considering the discriminant compounds of the comparison Lvs CTRL diets.**
(TIF)

**S1 Table. Feed component and proximate composition of the experimental diets.**
(XLSX)

**S2 Table. Lipidomics dataset resulting from the UHPLC-QTOF-MS analysis in both ESI + and ESI—ionisation modes, together with the discriminant compounds resulting from the comparisions L+P *vs.* CTRL, L *vs.* CTRL, and L *vs.* L+P when considering both univariate and multivariate statistical approaches.**
(XLSX)

**S3 Table. Genes differentially expressed in L *vs.* CTRL comparison, their average expression levels and their complete information.**
(XLSX)

**S4 Table. Results of the cluster analysis performed considering differentially expressed genes and discriminant lipid classes in L *vs*. CTRL.**
(XLSX)

**S5 Table. Results of the Principal Component Analysis (PCA) of the differentially expressed genes and discriminant lipid classes in L *vs*. CTRL.**
(XLSX)

**S6 Table. Results of the correlations between the genes and lipids contributing the most in differentiating L *vs*. CTRL.**
(XLSX)

**S7 Table. Genes differentially expressed in L+P *vs*. CTRL comparison, their average expression levels and their complete information.**
(XLSX)

**S8 Table. Results of the cluster analysis performed considering differentially expressed genes and discriminant lipid classes in L+P *vs*. CTRL.**
(XLSX)

**S9 Table. Results of the Principal Component Analysis (PCA) of the differentially expressed genes and discriminant lipid classes in L+P *vs*. CTRL.**
(XLSX)

**S10 Table. Results of the correlations between the genes and lipids contributing the most in differentiating L+P *vs*. CTRL.**
(XLSX)

**S11 Table. Genes differentially expressed in L *vs*. L+P comparison, their average expression levels and their complete information.**
(XLSX)

**S12 Table. Results of the cluster analysis performed considering differentially expressed genes and discriminant lipid classes in L *vs*. L+P.**
(XLSX)

**S13 Table. Results of the Principal Component Analysis (PCA) of the differentially expressed genes and discriminant lipid classes in L *vs*. L+P.**
(XLSX)

## Acknowledgments

The authors are deeply grateful to Giacinto Dalla Casa who took care of the animals with expertise and professionalism. The authors thank the "Romeo ed Enrica Invernizzi" foundation (Milan, Italy) for its kind support to the metabolomics facility at the Università Cattolica del Sacro Cuore.

## Disclaimer

Marika Vitali is currently employed with the European Food Safety Authority (EFSA) in the BIOHAW Unit that provides scientific and administrative support to EFSA's scientific activities in the area of Animal Health and Welfare. The positions and opinions presented in this article are those of the authors alone and are not intended to represent the views or scientific works of EFSA.

## Author Contributions

**Conceptualization:** Laura Righetti, Chiara Dall'Asta, Roberta Davoli, Gianni Galaverna.

**Data curation:** Gabriele Rocchetti, Marika Vitali, Martina Zappaterra, Laura Righetti, Rubina Sirri.

**Formal analysis:** Gabriele Rocchetti, Marika Vitali, Martina Zappaterra, Laura Righetti, Rubina Sirri.

**Funding acquisition:** Roberta Davoli, Gianni Galaverna.

**Investigation:** Gabriele Rocchetti, Marika Vitali, Martina Zappaterra.

**Methodology:** Gabriele Rocchetti, Marika Vitali, Martina Zappaterra, Laura Righetti.

**Resources:** Luigi Lucini, Chiara Dall'Asta, Roberta Davoli, Gianni Galaverna.

**Supervision:** Roberta Davoli, Gianni Galaverna.

**Visualization:** Gabriele Rocchetti.

**Writing – original draft:** Gabriele Rocchetti, Marika Vitali, Martina Zappaterra, Laura Righetti, Rubina Sirri.

**Writing – review & editing:** Luigi Lucini, Chiara Dall'Asta, Roberta Davoli, Gianni Galaverna.

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
