## [Decision Letter · Decision Letter 0]

2 Nov 2021

PONE-D-21-25228A molecular insight into the lipid remodelling of pig Longissimus thoracis muscle following dietary supplementation with functional ingredientsPLOS ONE

Dear Dr. Righetti,

Thank you for submitting your manuscript to PLOS ONE. After careful consideration, we feel that it has merit but does not fully meet PLOS ONE’s publication criteria as it currently stands. Therefore, we invite you to submit a revised version of the manuscript that addresses the points raised during the review process.

We look forward to receiving your revised manuscript.

Kind regards,

Marcio de Souza Duarte

Academic Editor

PLOS ONE

Journal Requirements:

Reviewers' comments:

Reviewer's Responses to Questions

**Comments to the Author**

1. Is the manuscript technically sound, and do the data support the conclusions?

Reviewer #1: Partly

2. Has the statistical analysis been performed appropriately and rigorously? 

Reviewer #1: I Don't Know

3. Have the authors made all data underlying the findings in their manuscript fully available?

Reviewer #1: Yes

4. Is the manuscript presented in an intelligible fashion and written in standard English?

Reviewer #1: Yes

5. Review Comments to the Author

Reviewer #1: General Comments:

This manuscript presents original results from a study evaluating longissimus muscle lipidome and transcriptome in swine fed diets with a different fat profile. Part of the data has already been published in two companion papers. The study is scientifically sound and has a good number of experimental units, but I think it is missing information about the carcass and meat traits. I mean, was there a difference in intramuscular fat?

The authors should review the proposed title. They did not evaluate lipid remodeling (i.e., synthesis and degradation). You have a transcriptomic snapshot and a lipidomic snapshot, but you did not evaluate it over time.

The language is good, although a couple of words could be replaced to improve clarity (please see the specific comments below).

General Questions:

1. Would authors consider renaming the treatments? It could make the text more informative [e.g., instead of D1 => CTRL; D2 = L; D3 => L+P].

2. Why do you choose to study meat fatty acid profile instead of subcutaneous FA profile? Maybe pork bellies FA.

3. Was there a difference in intramuscular fat among treatments?

4. I understand the contract D1 vs. D2, but why are you contrasting D1 vs D3, but not D2 vs D3? I mean how can you separate what is PUFA effect and what is the effect of polyphenols?

5. The introduction should be edited adding the rationality of adding polyphenol to pigs diet and why add it to a diet with PUFA.

6. Why did you run a supervised and unsupervised statistical model? I mean, shouldn’t you choose one only. If they show different results, which one are you going to consider?

Specific Comments:

L. 32- Is this quoted right? Does PUFA improve oxidative stability? If so, why do you have to add “natural antioxidants” as highlighted in L.33?

L. 37- I suggest replacing “determining the” by “leading to”

L. 48- Is it forage a source of fatty acids in swine’s diet?

L. 48- I suggest replacing “exploited” by “used”

L. 51- Please, replace the word “react”. I don’t think it is what you want to say

L. 51- Please review the citation “hormones8.”

L. 58- What are referring to with “later” here.

L. 65- Replace “pig meat” with “pork”

L. 93- 5% of what

L. 99- “slaughter”

L. 100- Please review across the manuscript the table and figure citations (e.g., L. 212, L.218). I think they should be capitalized

L. 103- 105 what?

L. 104-105- Please, specify the exact sampling place

L. 108- Delete “meat”

L. 108- “handshaking” is this term right?

L. 120- define “HRMS”

L. 153- “metabolomic” is this term right? (same for L.207)

L. 160- Why you did not evaluate D2vs. D3?

L. 162- I don’t think it is clear which “same software” you are talking about

L. 178-179- Please add a reference for these methods.

L. 198- At the beginning of the discussion, the authors should provide a little background on animal growth performance and carcass and meat quality traits. So, is this difference in lipidomic somehow affected by the pork fat percentage? I mean, we know that phospholipids have more PUFA than neutral lipids stored inside an adipocyte. So, is the fat percentage

L. 202- “lipid metabolites” is this term right?

L. 230- What are these “derivatives”? did you mention it earlier in the text?

L. 291- Can you add a reference to support this sentence?

L. 298- Please add a reference

L. 307-309- Did you evaluate fat oxidation? Like TBARS

L. 312-314- Please add a reference

L. 326- what is “CTR”

L. 333- add a “respectively” after “annotations”

L. 339- which “figure” are you talking about?

L. 354-358- I am not sure if your statistical comparison allows you to affirm this. In my opinion, you should contrast D2 and D3.

L. 364- what is “control group”?

L. 369- which “figure” are you talking about?

L. 388-391- I don’t think you can infer that PUFA is promoting lipid remodeling, based on your data. Again, lipid remodeling is a change over time, and you evaluated only one timepoint.

L. 394-397- Please add a reference

L. 410-412- See comment for L.388-391

Table 4- Which lipids are these genes correlated to? Total lipids in meat?

L. 540-541- Please rephrase this sentence.

L. 562- I don’t think ECM is considered “cytoskeleton”. Please review this sentence.

L. 571-572- Please add a reference

L. 583- I don’t think this approach is a good one to study cell membrane remodeling.

L. 585- Please rephrase it replacing “each experimental diet”. Here at the conclusion, you should show in a few sentences the effect of PUFA and polyphenol on lipidomic. Avoid using D1, D2, or D3.

L. 585- What do you mean by “restricted”?

Figure 1- I think you should relabel the axes. Maybe put PCA1 (% of the variance explained). So, here you are not using a consistent treatment label. I mean, there is no D1, D2, and D3.

Figure 4- how were these data generated? I mean did you contrast D2 and D3? A lipid that is more accumulated in D2 and D3 compared to D1 may not have a similar accumulation if you compare D2 against D3.

6. PLOS authors have the option to publish the peer review history of their article (what does this mean?). If published, this will include your full peer review and any attached files.

Reviewer #1: No

---

## [Author Response · Author response to Decision Letter 0]

15 Dec 2021

See "response to decision letter" attachment

---

## [Decision Letter · Decision Letter 1]

21 Feb 2022

A molecular insight into the lipid remodelling of pig Longissimus thoracis muscle following dietary supplementation with functional ingredients

PONE-D-21-25228R1

Dear Dr. Righetti,

We’re pleased to inform you that your manuscript has been judged scientifically suitable for publication and will be formally accepted for publication once it meets all outstanding technical requirements.

Kind regards,

Marcio Duarte, PhD

Academic Editor

PLOS ONE

Additional Editor Comments (optional):

Reviewers' comments:

Reviewer's Responses to Questions

**Comments to the Author**

1. If the authors have adequately addressed your comments raised in a previous round of review and you feel that this manuscript is now acceptable for publication, you may indicate that here to bypass the “Comments to the Author” section, enter your conflict of interest statement in the “Confidential to Editor” section, and submit your "Accept" recommendation.

Reviewer #1: All comments have been addressed

2. Is the manuscript technically sound, and do the data support the conclusions?

Reviewer #1: Yes

3. Has the statistical analysis been performed appropriately and rigorously? 

Reviewer #1: Yes

4. Have the authors made all data underlying the findings in their manuscript fully available?

Reviewer #1: Yes

5. Is the manuscript presented in an intelligible fashion and written in standard English?

Reviewer #1: Yes

6. Review Comments to the Author

Reviewer #1: The authors viewed and substantially improved the manuscript quality. I am satisfied with the modifications and answers to my questions.

7. PLOS authors have the option to publish the peer review history of their article (what does this mean?). If published, this will include your full peer review and any attached files.

Reviewer #1: No

---

## [Editor Report · Acceptance letter]

11 Mar 2022

PONE-D-21-25228R1 

A molecular insight into the lipid changes of pig *Longissimus thoracis* muscle following dietary supplementation with functional ingredients 

Dear Dr. Righetti:

I'm pleased to inform you that your manuscript has been deemed suitable for publication in PLOS ONE. Congratulations! Your manuscript is now with our production department. 

Kind regards, 

on behalf of

Dr. Marcio Duarte 

Academic Editor

PLOS ONE